# Uncertainty-aware Human Mobility Modeling and Anomaly Detection

## Abstract

Given the GPS coordinates of a large collection of human agents over time, how can we model their mobility behavior toward effective anomaly detection (e.g. for bad-actor or malicious behavior detection) without any labeled data? Human mobility and trajectory modeling have been studied extensively with varying capacity to handle complex input, and performance-efficiency trade-offs. With the arrival of more expressive models in machine learning, we attempt to model GPS data as a sequence of stay-point events, each with a set of characterizing spatiotemporal features, and leverage modern sequence models such as Transformers for un/self-supervised training and inference. Notably, driven by the inherent stochasticity of certain individuals' behavior, we equip our model with aleatoric/data uncertainty estimation. In addition, to handle data sparsity of a large variety of behaviors, we incorporate epistemic/model uncertainty into our model. Together, aleatoric and epistemic uncertainty enable a robust loss and training dynamics, as well as uncertainty-aware decision making in anomaly scoring. Experiments on large expert-simulated datasets with tens of thousands of agents demonstrate the effectiveness of our model against both forecasting and anomaly detection baselines. All code is available at `https://anonymous.4open.science/r/mobility-ad`.

## 1 Introduction

Anomaly detection of human mobility has become crucial for various applications, ranging from security and surveillance to health monitoring. Accurately detecting anomalous human behavior from GPS data can reveal critical insights, such as abnormal patterns that indicate security threats or spread of infectious diseases Meloni et al. (2011); Barbosa et al. (2018); Stanford et al. (2024).

However, the complexity and inherent uncertainty of human behavior make this task particularly challenging: (1) The first difficulty lies in capturing the complex *spatiotemporal dependencies* both within individual activities and between multiple activities across different times and locations. The former aims to capture the correlations between different spatiotemporal features (a.k.a. markers) of a single activity; such as the time-of-day (e.g. 4am) being indicative of POI location (e.g. home). The latter associates with the dependency patterns between different activities over time (e.g. restaurant at weekday lunchtime followed by office building). (2) Furthermore, human activity data is abound with *uncertainty*—arising from the unpredictable behavior of inherently stochastic individuals as well as data sparsity—making accurate anomaly detection even more difficult.

Prior work on behavior modeling often focused on temporal event forecasting, either ignoring spatial information Minor et al. (2015); Manzoor and Akoglu (2017) or based only on a few markers Du et al. (2016); Zhou et al. (2022). With the emergence of the Transformer architecture Vaswani et al. (2023), more recent work modeled complex spatiotemporal events more expressively Xue et al. (2022); Corrias et al. (2023). However, to our knowledge, no existing work considered uncertainty-aware anomaly detection of human mobility.

To address both of the above challenges, we introduce UIFORMER for uncertainty-incorporated human behavior modeling and anomaly detection, which takes advantage of a "dual" Transformer architecture Truong Jr and Bepler (2023) as well as aleatoric and epistemic uncertainty into account. Transformer has emerged as the de facto model for modern AI problems, due to its superior ability to model long-range dependencies. Through a dual architecture equipped with both feature- and event-level attention, we address the first challenge in capturing the intra- and inter-event dependencies.

For the second challenge, we first account for data (a.k.a. aleatoric) uncertainty, which stems from the inherent stochasticity in certain human behavior. For example, it is typical of some populations (e.g. retired individuals) to be less predictable relative to some others (e.g. shift workers with a strict schedule). In addition to data uncertainty, we incorporate model (a.k.a. epistemic) uncertainty, which typically stems from incomplete or sparse observations from a potentially complex underlying data generating distribution. Explicitly modeling both aleatoric (data) and epistemic (model) uncertainty benefits UIFORMER in two ways: First, it factors into the learning objective, enabling a more robust training. Second, it allows the model to reason effectively about the confidence of its predictions, thus enabling more reliable inference. Specifically, while it is typical for forecasting-based detection methods to flag anomalies based on the deviation between the predicted and observed values alone, our uncertainty-aware UIFORMER considers both the predicted values as well as total estimated uncertainties, thus providing a more nuanced and accurate approach to anomaly detection.

The following summarizes the main contributions of this work.

- **Problem formulation:** We cast the anomaly detection problem in human mobility onto sequence prediction, where we translate the regular GPS readings (of the form $\langle x, y, t \rangle$) into irregular stay-point events over time with spatiotemporal features. This step resembles a form of "tokenization" of the raw input data to a higher-level representation for sequence modeling.
- **Uncertainty-aware dependency modeling:** We design UIFORMER, a *dual* Transformer architecture that is *uncertainty-aware*; which is equipped with ($i$) both feature-level and event-level attention—for capturing dependencies among features as well as across events, respectively; and ($ii$) both data and model uncertainty estimation—toward accounting for inherent stochasticity in human behavior as well as data scarcity, respectively.
- **Uncertainty-aware anomaly scoring:** Beyond prediction, we extend our uncertainty modeling to anomaly detection, which allows the model to flag abnormal behavior with greater accuracy by factoring in both prediction value and uncertainty estimates.
- **Effectiveness:** Through extensive experiments on expert-simulated benchmark datasets, we show that UIFORMER significantly outperforms both forecasting as well as anomaly detection baselines. Ablation analyses demonstrate UIFORMER's key advantages: accurate uncertainty estimates and effective uncertainty-aware anomaly scores.

## 2 PROBLEM AND PRELIMINARIES

We introduce related concepts and formulate the anomaly detection problem on human mobility.

**Stay-point Event.** As shown in Figure 1, a stay-point event is extracted from raw GPS data that is used to describe an individual's daily activity. Let $e_i$ be an event with spatiotemporal features (or markers) $\boldsymbol{x}_i$:

$$\boldsymbol{x}_i = (x_i^{\text{st}}, x_i^{\text{sd}}, x_i^{\text{x}}, x_i^{\text{y}}, x_i^{\text{poi}}, x_i^{\text{dow}}), \quad (1)$$

where $\mathcal{F}_n = \{x_i^{\text{st}}, x_i^{\text{sd}}, x_i^{\text{x}}, x_i^{\text{y}}\}$ is the numerical feature set: $x_i^{\text{st}}$ and $x_i^{\text{sd}}$ are the start time and stay duration of the event, $x_i^{\text{x}}, x_i^{\text{y}}$ are the two-dimensional coordinates depicting the latitude and longitude of the event's location. $\mathcal{F}_c = \{x_i^{\text{poi}}, x_i^{\text{dow}}\}$ is the categorical feature set, depicting the Point-of-Interest (POI) such as office building, store, etc. and Day-of-Week (DOW) of the event, respectively.

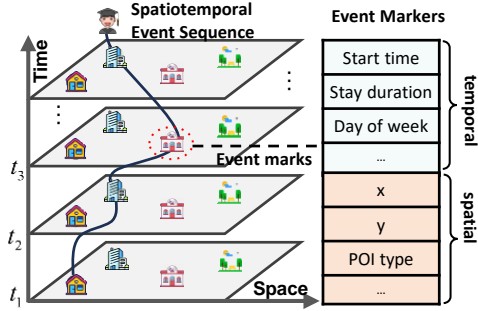

Figure 1: Spatiotemporal event sequence with various features (a.k.a. markers) per event.

**Spatiotemporal Event Sequence.** Let $\mathcal{E}^u = [e_1^u, e_2^u, \ldots, e_{N_u}^u]$ be the event sequence that records all historical events of individual $u$ in the dataset, where $N_u$ is $u$'s total number of historical events. Furthermore, let $e_i^{u,d}$ be the $i$-th event of individual $u$ at day $d$, and $N_{u,d}$ be the number of events of $u$ on day $d$. The event sequence with $w$-day time window can be given as

$$\mathcal{E}_w^u = [e_1^{u,d-w}, e_2^{u,d-w}, \ldots, e_1^{u,d-w+1}, \ldots, e_1^{u,d}, \ldots, e_{N_{u,d}}^{u,d}]. \quad (2)$$

**Unsupervised Human Mobility Anomaly Detection.** Given a individual $u$'s event sequence in a $w$-day window $\mathcal{E}_u^w$ and a target event $e \in \mathcal{E}_u^w$, we aim to learn an anomaly score function to identify whether $e$ is anomalous, i.e. not aligned with $u$'s typical behavior pattern, without any labeled data.

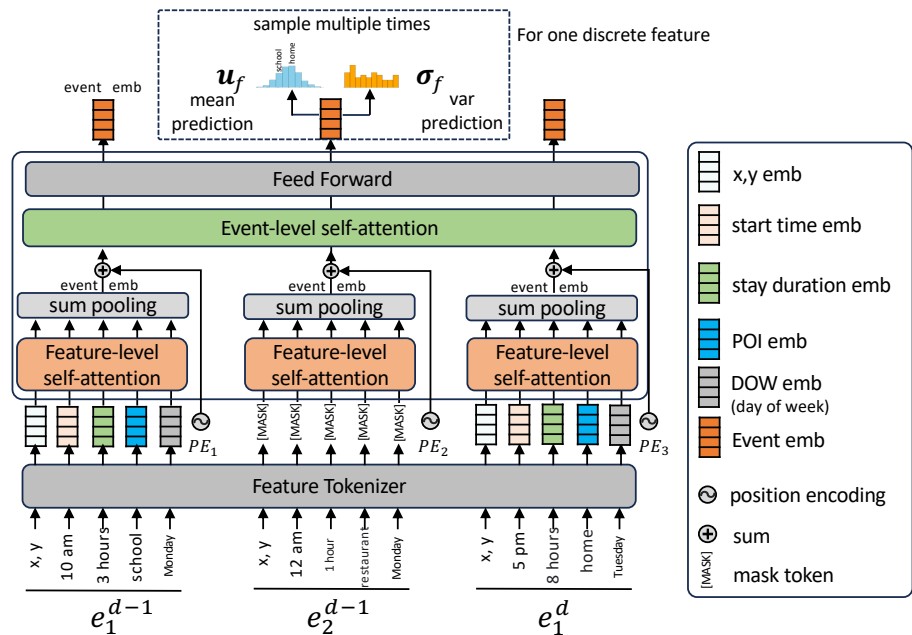

Figure 2: Proposed model architecture for human mobility behavior modeling. Raw GPS data is represented as a (ordered) sequence of stay-point events each with a (unordered) sequence of spatiotemporal markers. Dual-Transformer architecture models each individual's sequence-of-sequences via both feature- and event-level attention, as well as uncertainty estimation that simultaneously enables robust training and inform anomaly scoring at inference.

## 3 PROPOSED MODEL

Figure 2 illustrates the overall architecture of UIFORMER, which learns human mobility patterns in a self-supervised manner through pre-training, with the goal of accurately reconstructing masked events as well as evaluating the prediction uncertainty. In a nutshell, the feature tokenizer first projects features into high-dimensional space by representing each event's feature as an individual token (§3.1). Then, the Dual-Transformer encoder further encodes the input by both feature-level and event-level self-attention (§3.2). Lastly, unlike previous work, we introduce the uncertainty-aware decoder to recover all features of the masked event along with the aleatoric and epistemic uncertainty associated with each feature (§3.3). These estimates are used to compare the predicted versus observed events at inference time for uncertainty-incorporated anomaly scoring (§3.4).

### 3.1 FEATURE TOKENIZER

The feature tokenizer transforms all features $\boldsymbol{x} \in \mathbb{R}^F$ of an event into embeddings $\boldsymbol{e} \in \mathbb{R}^D$, where $F$ is the number of input features and $D$ is the embedding dimension. Specifically, a numerical feature $x_j^{(num)}$ is projected by a linear transformation with weight $\boldsymbol{W}_j^{(num)} \in \mathbb{R}^D$ and bias $\boldsymbol{b}_j \in \mathbb{R}^D$, and the embedding of a categorical feature is implemented as the embedding lookup table $\boldsymbol{W}_j^{(cat)} \in \mathbb{R}^{C_j \times D}$, where $C_j$ is the total number of categories for feature $x_j^{(cat)}$. Overall:

$$\boldsymbol{e}_j^{(num)} = \boldsymbol{b}_j^{(num)} + x_j^{(num)} \cdot \boldsymbol{W}_j^{(num)} \qquad \in \mathbb{R}^D \ ,$$

$$\boldsymbol{e}_j^{(cat)} = x_j^{(cat)} \boldsymbol{W}_j^{(cat)} \qquad \in \mathbb{R}^D \ ,$$

$$\boldsymbol{e} = \texttt{stack} \left[ \boldsymbol{e}_1^{(num)}, \ \dots, \ \boldsymbol{e}_{k^{(num)}}^{(num)}, \ \boldsymbol{e}_1^{(cat)}, \ \dots, \ \boldsymbol{e}_{k^{(cat)}}^{(cat)} \right] \in \mathbb{R}^{F \times D} \ .$$

where $x_j^{(cat)}$ is represented as a one-hot vector. The feature embedding $\boldsymbol{e} \in \mathbb{R}^{F \times D}$ of one event is the concatenation of all numerical embeddings and categorical embeddings. Denoting $B$ as the batch size and $L$ the max length of $\mathcal{E}_w^u$ in a batch, the output of the feature tokenizer is $\boldsymbol{E}_0 \in \mathbb{R}^{B \times L \times F \times D}$.

## 3.2 DUAL TRANSFORMER ENCODER

Inspired by (Truong Jr and Bepler, 2023), we design a Dual Transformer encoder that treats the data as a sequence-of-sequences, which takes the sequence of events each with a sequence of feature embeddings as input, and models both feature-level and event-level interactions with two types of Transformer-based (Vaswani et al., 2023) components. A description of the Transformer block is included in Appendix B.

**Feature-level Transformer.** By considering each feature as an input token to the Transformer, the Feature-level Transformer represents a feature in a way that explicitly incorporates other features' information. In the implementation, we convert the feature tokenizer's output $E_0 \in \mathbb{R}^{B \times L \times F \times D}$ into $E_1 \in \mathbb{R}^{(B \times L) \times F \times D}$, which is then fed into $M_1$ Transformer blocks to get the updated feature embeddings $E_2 \in \mathbb{R}^{(B \times L) \times F \times D}$. Note that we do not include any positional embedding as input for this module as there is no sequential relationship between the features of an event.

**Event-level Transformer.** Based on $E_2$, we calculate an event's embedding by averaging all its corresponding feature embeddings, which results in the input of the Event-level Transformer $E_3 \in \mathbb{R}^{B \times L \times D}$. The Event-level Transformer then captures the dependencies (e.g., sequential patterns) between different events by considering each event as a token. Moreover, considering that the sequential information in the event sequence is important to reflect human mobility behavior, we design two types of positional encoding for each event: ($i$) Sequence positional encoding; used to describe the order of an event in the input sequence, and ($ii$) Within-day positional encoding (as shown in Figure 3); used to describe the order of the event in its specific day, explicitly demarcating day boundaries. After $M_2$ updates of the Event-level Transformer blocks, we get the final output of the encoder $\bar{E} \in \mathbb{R}^{B \times L \times D}$.

**Within-day Position Encoding**

| Day $d_1$: | $e_1^{d_1}$ | $e_2^{d_1}$ | $e_3^{d_1}$ | $e_4^{d_1}$ |
| Day $d_2$: | $e_1^{d_2}$ | $e_2^{d_2}$ | $e_3^{d_2}$ | |
| | **0** | **1** | **2** | **3** |

Figure 3: Within-day position encoding.

## 3.3 UNCERTAINTY-AWARE DECODER

**Motivation.** Uncertainty modeling has shown great promise in various fields, such as computer vision (Kendall and Gal, 2017; Kendall et al., 2016) and natural language processing (Xiao and Wang, 2019; Gal and Ghahramani, 2016b). In particular, human mobility sequences are inherently associated with uncertainty. For instance, certain individuals (e.g., shift workers) may exhibit highly predictable patterns (i.e., low uncertainty), while others (e.g., retirees) can be much more unpredictable in their activities over time. This type of uncertainty, known as aleatoric uncertainty (or data uncertainty) (Kendall and Gal, 2017), arises naturally from the data itself and cannot be reduced by simply adding more training data. In addition to aleatoric uncertainty, our setting also involves epistemic uncertainty, which refers to the uncertainty in the model itself due to limited knowledge. This type of uncertainty captures the model's confidence in its own predictions and can be reduced by acquiring more data or improving the model specification or architecture. In human event sequence modeling, epistemic uncertainty is crucial for identifying situations where the model lacks sufficient information to make confident predictions, especially in underrepresented or complex scenarios. Furthermore, given that different features can have different predictability even for the same individual/event, we design a feature-level uncertainty learning mechanism. In the following part, we elaborate on how we model the epistemic and aleatoric uncertainty for both numerical and categorical features.

### 3.3.1 EPISTEMIC UNCERTAINTY MODELING

Inspired by (Kendall, 2018), we model the epistemic uncertainty by placing a prior distribution over the model's parameters, and then try to capture how much these parameters vary given observed data.

To ease the presentation, we use $s$ to denote the input instance $\mathcal{E}_w^u$, and $\theta$ to denote all parameters of the model $\mathcal{M}$. We start by assuming $\theta$ follows a prior distribution, which reflects our initial belief about the possible values of these parameters. The goal is to update this belief based on the available data $s$ by calculating the posterior distribution of $\theta$, that is $p(\theta|s)$. This posterior captures how the parameters might vary, thereby reflecting the model's uncertainty about its predictions. Formally, Bayes' theorem (Bayes, 1763) gives us the posterior: $p(\theta|s) = \frac{p(s|\theta)p(\theta)}{p(s)}$.

However, computing the exact posterior distribution in deep learning models is often intractable due to the high dimensionality of the parameter space and the complexity of the model. As a practical alternative, we apply Monte Carlo (MC) Dropout (Gal and Ghahramani, 2016a) to approximate the posterior. In practice, we introduce dropout layers after the embedding layer of each feature with a dropout ratio of 0.05, which randomly drops a subset of neurons during both training and inference. MC Dropout works by sampling different subsets of model parameters $\theta$ during each forward pass, thereby creating an ensemble of models. The variance in the predictions across these passes provides an estimate of the epistemic uncertainty.

*Epistemic uncertainty for numerical features:* The decoding of numerical feature $f \in \mathcal{F}_n$ is modeled as a regression task. In this case, the variance of the predictions across $T$ (equals 50 in our model) steps of stochastic forward passes quantifies the epistemic uncertainty $\alpha_f^{num}$ for $f$, formulated as

$$\alpha_f^{num} : \text{Var}(y) = \frac{1}{T} \sum_{t=1}^{T} \left( \mathcal{M}_{\theta_t}(s) - \bar{y} \right)^2 \ , \quad \text{where } \bar{y} = \frac{1}{T} \sum_{t=1}^{T} \mathcal{M}_{\theta_t}(s) \ , \tag{3}$$

*Epistemic uncertainty for categorical features:* The decoding of categorical feature $f \in \mathcal{F}_c$ is modeled as a classification task. We apply the softmax function to the logits produced by the model, giving a probability distribution over the classes: $p(y|\mathcal{M}_\theta(s)) = \text{softmax}(\mathcal{M}_\theta(s))$. Here, the epistemic uncertainty is measured by the entropy of the predicted probability distributions averaged by multiple forward processes, formulated as

$$\alpha_f^{cls} : H(p) = -\sum_c p_c \log(p_c) \ , \quad \text{where } p(y=c|s) = \frac{1}{T} \sum_{t=1}^{T} \text{softmax}(\mathcal{M}_{\theta_t}(s)) \ . \tag{4}$$

Overall, we approximate the epistemic uncertainty for both numerical and categorical features by MC Dropout, providing a practical and scalable way to capture the model's uncertainty in its predictions.

### 3.3.2 ALEATORIC UNCERTAINTY MODELING

We model aleatoric uncertainty by placing a distribution over the model's output. For example, in this paper, we model the regression output as a Gaussian distribution with random noise, i.e., $p(y|\mathcal{M}_\theta(s)) = \mathcal{N}(\mathcal{M}_\theta(s), \sigma^2)$, where aleatoric uncertainty aims to learn the variance of noise as a function of the input data.

*Aleatoric uncertainty for numerical features:* Let $\sigma_f$ denote the noise level of a feature $f \in \mathcal{F}_n$, which captures how much noise we have in the output of the feature. Moreover, we assume that $\sigma_f$ is data-dependent and can vary with the input. Then, $\sigma_f$ can be learned as a function of the data:

$$\mathcal{L}_f^{reg} = \frac{1}{2\sigma_f^2} \|y_f - \widehat{y}_f\|^2 + \frac{1}{2} \log \sigma_f^2 \ , \tag{5}$$

where $y_f$ and $\widehat{y}_f$ are the true and predicted value of $f$. $\sigma_f$ is also a model's output that serves as a learned loss attenuation. It gives less penalty to the model when the input data is associated with high aleatoric uncertainty, thus making the loss more robust to noisy data. The second term in Eq. (5) is a regularization term that prevents the model from learning a high uncertainty score for all instances, trivially driving the first term to zero. In practice, we further adopt a more numerically stable loss given as: $\mathcal{L}_f^{reg} = \frac{1}{2} \exp(-r_f) \|y_f - \hat{y}_f\|^2 + \frac{1}{2} r_f$, where $r_f = \log \sigma_f^2$. Instead of predicting the variance directly, predicting the log variance followed by an exponential function can ensure the positive value of the variance and make the training more numerically stable. In that case, the aleatoric uncertainty for numerical feature $f$ is denoted as $\beta_f^{num} := \sigma_f^2$.

*Aleatoric uncertainty for categorical features:* Let $C_f$ denote the number of unique categories (i.e. classes) of feature $f \in \mathcal{F}_c$. For the classification task, the model assumes that the prediction logits at sample time $t$ for each category follows a Gaussian distribution:

$$\widehat{\boldsymbol{m}}_{f,t} = \boldsymbol{u}_f + \boldsymbol{\sigma}_f \boldsymbol{\epsilon}_t, \quad \boldsymbol{\epsilon}_t \sim \mathcal{N}(0, I) \ , \tag{6}$$

where the predicted mean logits $\boldsymbol{u}_f \in \mathbb{R}^{C_f}$ and uncertainty term $\boldsymbol{\sigma}_f \in \mathbb{R}^{C_f}$ are the model outputs, as a function of its parameters at the $t$-th sample time. $\boldsymbol{\epsilon}_t \in \mathbb{R}^{C_f}$ represents a random vector drawn from a unit normal distribution $\mathcal{N}(0, I)$. Here $\boldsymbol{\sigma}_f$ accounts for the inherent uncertainty in the feature when conducting the prediction. The aleatoric uncertainty of categorical feature $f$ is then the average of variance across all classes, given as $\beta_f^{cls} := \frac{1}{C_f} \sum_{c'} \boldsymbol{\sigma}_{c'}^2$.

To learn such uncertainty, the loss function $\mathcal{L}_f^{cls}$ calculates the cross entropy based on the average predicted logits, normalized over all possible categorical outcomes:

$$\mathcal{L}_f^{cls} = -\log \frac{1}{T} \sum_t \exp \left( \widehat{m}_{f,t,c} - \log \sum_{c'} \exp \widehat{m}_{f,t,c'} \right). \qquad (7)$$

The above loss function achieves a similar effect to the numerical case, which can also be considered as learning the loss attenuation (Kendall and Gal, 2017). When the model assigns a high logit value $m_c$ to the observed class $c$, and the noise value $\sigma_c$ is low, the loss approaches zero — which is the desired outcome. Overall, the total training loss is the sum of the regression and classification losses across all features, given as:

$$\mathcal{L} = \sum_{f \in \mathcal{F}_n} \mathcal{L}_f^{reg} + \lambda \sum_{f \in \mathcal{F}_c} \mathcal{L}_f^{cls}, \qquad (8)$$

where $\lambda$ is a hyper-parameter designated to balance the scale of regression and classification losses. In our model, we set $\lambda = 1$ for simplicity.

### 3.4 Uncertainty-incorporated Anomaly Scoring

We compute event-level anomaly scores by aggregating deviations between the predicted and the observed values of all features, incorporating both epistemic and aleatoric uncertainties into the computation. For each test event on a given day, we first create a sequence of events spanning a 3-day window, including the previous and next day as context. We mask the event of interest and use the pre-trained UIFORMER to predict the masked features. For numerical features, we calculate the deviation as: $\Delta_f = |y_f - \widehat{y}_f|$ with $y_f$ and $\widehat{y}_f$ depicting the observed and predicted values, respectively. For categorical features, the deviation is computed as: $\Delta_f = 1 - \widehat{p}_f$ where $\widehat{p}_f$ is the predicted probability of the observed class. UIFORMER learns the inherent patterns of human behaviors using only normal data during training. Intuitively, higher deviations from these patterns suggest that an event is less consistent with learned behaviors and thus, more anomalous. Then, we can compute an anomaly score (AS) for a target event $e$ from the deviations alone as:

$$\text{AS}(e) = \text{agg} \left( \text{pth} \left( \Delta_f \right) \mid f \in \mathcal{F}_n \cup \mathcal{F}_c \right), \qquad (9)$$

where $\text{agg}(\cdot)$ denotes the aggregation function (e.g., mean or max) applied over $\Delta_f$'s across all features $\mathcal{F}$. The percentile function $\text{pth}(\cdot)$ is used to standardize deviations across features of different scales, effectively capturing the relative ranking of the deviation for each feature. This helps ensure that features with smaller absolute deviations are not undervalued compared to those with larger deviations. However, using deviations alone to detect anomalies is insufficient due to the diverse nature of human mobility. Some agents may deviate from the general population's behavior but are not necessarily anomalous. For instance, a (stochastic) agent with a tendency to explore different locations frequently (i.e., with high data uncertainty) may have higher deviations naturally, which does not imply an anomaly. To account for such cases, we propose to incorporate uncertainty estimates into the anomaly score. By scaling down deviations by the uncertainties, we can prevent inflating the anomaly scores of agents whose behaviors, though atypical, are still consistent with natural variability. As such, we compute an uncertainty-incorporated anomaly score (UI-AS) for a target event $e$ as:

$$\text{UI-AS}(e) = \text{agg} \left( \text{pth} \left( \frac{\Delta_f}{1 + \alpha_f + \beta_f} \right) \mid f \in \mathcal{F}_n \cup \mathcal{F}_c \right), \qquad (10)$$

where $\alpha_f + \beta_f$ denotes the total of the aleatoric and epistemic uncertainties for feature $f$.

## 4 Experiments

We conduct extensive experiments to investigate the following research questions (RQs):
**RQ1:** What is the performance of UIFORMER in the masked prediction task?
**RQ2:** Can our model outperform previous approaches in the anomaly detection task?
**RQ3:** Does UIFORMER have the ability to accurately capture aleatoric and epistemic uncertainty?
**RQ4:** How does each uncertainty component contribute to UIFORMER?

**Datasets.** We utilize two human activity datasets, namely SimLA and NUMOSIM. SimLA is an expert-simulated dataset that contains human mobility data in greater Los Angeles area. NUMOSIM (Stanford et al., 2024) is a publicly-available synthetic human mobility dataset, designed to benchmark anomaly

detection techniques for mobility data. Both datasets are obtained by training deep learning models on real mobility data and simulating realistic human mobility patterns, incorporating both normal and anomalous behaviors. The statistics of the two datasets are shown in Appendix Table 5.

We split the data into train/validation/test temporally by date, respectively spanning 3/1/2 weeks. During training, we randomly mask events in each sequence using a specified mask ratio. For validation and test data, we mask only one event per sequence, aligning with how we use the pre-trained model for anomaly detection.

## 4.1 RQ1: Masked Prediction Results

**Settings.** For baselines, we include classical deep models MLP (Murtagh, 1991), LSTM (Hochreiter and Schmidhuber, 1997), the original Transformer (Vaswani et al., 2023), as well as Dual-Tr; a variant of UIFORMER that removes the uncertainty estimation mechanism and only keeps the dual Transformer architecture (w/ cross-feature and cross-event attention). In addition, since UIFORMER can estimate the uncertainty of its prediction, we also report its conformal prediction performance where the top-5% most uncertain test samples are not evaluated, denoted by UIFORMER (5%). We report MAE and MAPE to evaluate the prediction performance of numerical features and use accuracy (ACC) for the categorical features. Higher ACC and lower MAE and MAPE indicate better performance. The models are trained on 1 GPU of NVIDIA RTX A6000. We search the hyperparameters in a given model space for all deep models, and each model's best performance is reported in the experiments. Detailed experiment settings and model configurations can be found in Appendix C.

Table 1: Masked prediction results. Uncertainty-aware UIFORMER outperforms the baselines, while allowing conformal prediction. Abbr: ST; start time (min), SD; stay duration (min), x, y; lat/lon (km).

| Method | SimLA | | | | | | | | | NUMOSIM | | | | | | | | |
| | MAE | | | | MAPE (%) | | | | ACC(%) | MAE | | | | MAPE (%) | | | | ACC(%) |
| | x | y | ST | SD | x | y | ST | SD | POI | x | y | ST | SD | x | y | ST | SD | POI |
|---|---|---|---|---|---|---|---|---|---|---|---|---|---|---|---|---|---|---|
| MLP | 9.73 | 10.16 | 208.07 | 420.92 | 188.13 | 500.59 | 43.28 | 569.01 | 45.61 | 15.97 | 15.74 | 207.64 | 396.58 | 416.55 | 361.58 | 339.15 | 572.69 | 56.08 |
| LSTM | 4.73 | 4.63 | 19.85 | 322.62 | 204.73 | 195.76 | 18.25 | 420.76 | 63.32 | 6.92 | 6.40 | 34.41 | 343.82 | 265.52 | 139.71 | 212.30 | 547.02 | 68.29 |
| Transformer | 2.67 | 2.65 | 26.85 | 77.12 | 132.33 | 115.35 | 17.95 | 66.82 | 73.49 | 4.08 | 3.45 | 21.57 | 71.33 | 143.81 | 72.15 | 15.19 | 68.51 | 75.90 |
| Dual-Tr | 2.71 | 2.63 | 20.62 | 76.53 | 131.72 | 117.29 | 16.41 | 51.95 | 73.74 | 3.63 | 3.48 | 21.69 | 80.08 | 134.32 | 71.34 | 17.36 | 60.17 | 76.12 |
| UIFORMER | 2.44 | 2.29 | 11.42 | 35.77 | 109.44 | 102.47 | 15.86 | 20.93 | 72.97 | 3.68 | 3.60 | 26.18 | 45.25 | 121.33 | 84.01 | 236.47 | 28.87 | 74.33 |
| UIFORMER (5%) | 2.05 | 1.91 | 8.65 | 21.98 | 99.39 | 89.45 | 1.18 | 20.66 | 75.58 | 3.31 | 3.21 | 8.30 | 33.61 | 109.94 | 74.45 | 1.23 | 28.86 | 76.95 |

**Results.** Table 1 shows the masked prediction results (here and after, we bold the **best** performance and underline the second best). UIFORMER delivers the best performance across both datasets and for all features compared with the baselines. In SimLA, UIFORMER achieves the lowest MAE for start time (ST) and stay duration (SD) with values of 11.42 and 35.77 minutes, respectively, outperforming the powerful sequence model, Transformer, which yields 26.85 for ST and 77.12 for SD. Compared to Dual-Tr, building uncertainty into UIFORMER leads to significant improvements in prediction performance; in SimLA, UIFORMER reduces MAE for ST from 20.62 to 11.42 minutes, and MAPE for SD from 51.95% to 21.93%, underscoring the significant impact of modeling both epistemic and aleatoric uncertainty in achieving more accurate and robust predictions. Lastly, UIFORMER (5%) further improves over UIFORMER, which demonstrates UIFORMER's ability to identify highly uncertain samples and that it can be quite accurate on its certain predictions.

## 4.2 RQ2: Anomaly Detection Results

**Settings.** There are only a few related public achievements for human mobility anomaly detection, thus it is difficult to find state-of-the-art models for direct comparison. For a comprehensive evaluation, we implement some baselines as well as extract core components from related literature in trajectory anomaly detection, including Spatial-Temporal Outlier Detector (STOD) (Cruz and Barbosa, 2020), RioBus (Bessa et al., 2016), a self-supervised (SS) MLP, LSTM, and Transformer. STOD and RioBus are designed for bus trajectories to predict the Bus ID from the sequence of GPS coordinates which we repurpose for human trajectories. SS-MLP is trained to distinguish real events from fake ones that are synthetically-constructed via random alterations. Further details on baselines are given in Appendix D. We report two common metrics for evaluation: AUROC (Area Under ROC curve) and AUPR (Area Under Precision-Recall curve). These are frequently used for anomaly detection as quantities of ranking quality. As the percentage of anomalous events is very small on the original datasets, we construct two additional one with around 5% anomalies by including agents with anomalous events

and randomly sampling normal agents without any anomalous events. Dataset statistics for anomaly detection can be found in Appendix Table 6.

Table 2: Anomaly detection results at the event-level (left) and the agent-level (right).

| Method | Event-level Anomaly Detection | | | | | | Agent-level Anomaly Detection | | | | | |
| --- | --- | --- | --- | --- | --- | --- | --- | --- | --- | --- | --- | --- |
| | SimLA (small) | | NUMOSIM (small) | | NUMOSIM (20K) | | SimLA (small) | | NUMOSIM (small) | | NUMOSIM (20K) | |
| | AUROC | AUPR | AUROC | AUPR | AUROC | AUPR | AUROC | AUPR | AUROC | AUPR | AUROC | AUPR |
| STOD | 0.577 | 0.073 | 0.571 | 0.071 | 0.584 | 0.003 | 0.506 | 0.655 | 0.511 | 0.468 | 0.519 | 0.020 |
| RioBus | 0.503 | 0.044 | 0.530 | 0.047 | 0.532 | 0.002 | 0.488 | 0.632 | 0.510 | 0.453 | 0.508 | 0.019 |
| SS-MLP | 0.596 | 0.172 | 0.630 | 0.144 | 0.634 | 0.008 | 0.653 | 0.777 | **0.697** | **0.638** | 0.703 | 0.053 |
| LSTM | 0.755 | 0.182 | 0.637 | 0.122 | 0.642 | 0.020 | 0.631 | 0.706 | 0.599 | 0.526 | 0.658 | 0.062 |
| Transformer | 0.723 | 0.232 | 0.635 | 0.132 | 0.636 | 0.033 | 0.679 | 0.785 | 0.625 | 0.525 | 0.717 | 0.095 |
| UIFORMER | **0.795** | **0.329** | **0.694** | **0.146** | **0.670** | **0.034** | **0.746** | **0.841** | 0.655 | 0.572 | **0.727** | **0.114** |

**Results.** Table 2 presents the anomaly detection results for both event-level and agent-level anomalies. UIFORMER consistently outperforms the baselines across all datasets in detecting anomalous events and agents. At the event level, particularly on SimLA, UIFORMER achieves a significantly large margin against the baselines, with the highest AUROC of 0.795 and AUPR of 0.329 (8× lift over the 0.041 base rate). This highlights the effectiveness of incorporating both aleatoric and epistemic uncertainty, which substantially enhances event-level detection. UIFORMER also performs well for agent-level anomaly detection (w/ maximum anomaly scores across all events per agent). On the NUMOSIM (20K agents) dataset, it achieves an AUROC of 0.727 and an AUPR of 0.114 (57× lift over the 0.002 base rate), demonstrating its ability to effectively identify rare anomalous agents within a large population. By incorporating uncertainty into the anomaly score computation, UIFORMER prevents inflated scores for agents with exploratory but not necessarily anomalous behavior, thereby reducing false positives. As a result, it more effectively detects true anomalies that deviate from otherwise predictable behavior without incorrectly flagging exploratory behavior.

### 4.3 RQ3: ANALYSIS OF UNCERTAINTY ESTIMATION

Next we scrutinize the quality of the estimated uncertainty scores. To this end, we analyze the relationship between prediction accuracy as a function of total uncertainty, as well as how the estimated aleatoric and epistemic uncertainty scores vary by the amount of training data using SimLA.

**Relationship between uncertainty and MAE/accuracy.** We first sort the test samples by the total AU+EU uncertainty score (i.e., $\sum_{f \in \mathcal{F}} \alpha_f + \beta_f$), and then gradually remove test samples with uncertainty larger than a certain percentile threshold while recording the prediction performance of UIFORMER on the rest. As shown in Figure 4, UIFORMER's performance gradually improves as we exclude test samples with large total uncertainty. This means that the uncertainty estimates are well aligned with prediction performance, as all MAE (Accuracy) curves follow a monotonically increasing (decreasing) trend for numerical (categorical) features. Notably, for stay duration (SD) prediction (middle), MAE increases sharply from around 20 minutes to 35 minutes due to the top 5% most uncertain samples. As for the POI prediction (right), around 50% of the most certain samples can yield a quite promising performance that achieves near 100% accuracy, which gradually drops with increasing uncertainty. Interestingly, AU-only UIFORMER (in blue) appears to be more reflective of these expected trends for x and SD, while it is the EU-only variant (in purple) for POI, demonstrating the complementary benefit of estimating these different uncertainty types.

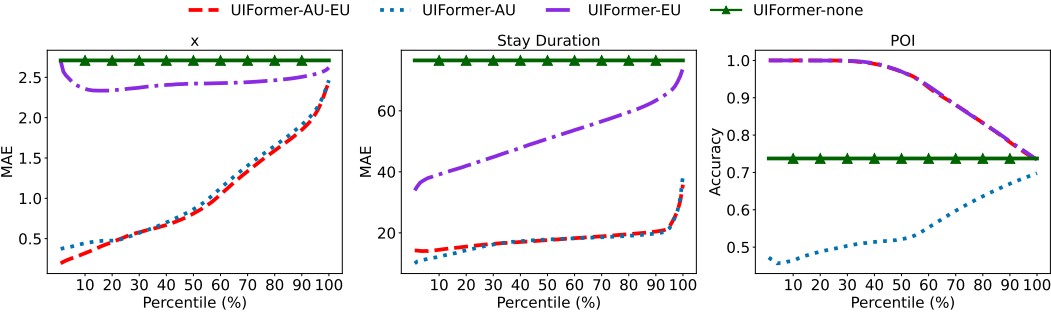

Figure 4: Relationship btwn. MAE (accuracy) & uncertainty follow an increasing (decreasing) trend.

Table 3: Aleatoric (AU) and epistemic uncertainties (EU) under different training data size settings.

| Train | Test | x | | y | | Start Time | | Stay Duration | | POI | |
|---|---|---|---|---|---|---|---|---|---|---|---|
| | | AU $\times 10^{-3}$ | EU $\times 10^{-5}$ | AU $\times 10^{-3}$ | EU $\times 10^{-5}$ | AU $\times 10^{-3}$ | EU $\times 10^{-5}$ | AU $\times 10^{-3}$ | EU $\times 10^{-5}$ | AU $\times 10^{-3}$ | EU |
| 1-week | 2-week | 1.30 | 2.37 | 2.40 | 4.00 | 0.85 | 15.49 | 0.10 | 0.48 | 17.11 | 1.01 |
| 2-week | 2-week | 1.44 | 2.05 | 2.45 | 3.12 | 0.61 | 7.29 | 0.13 | 0.25 | 14.34 | 0.98 |
| 3-week | 2-week | 1.36 | 1.23 | 2.51 | 2.00 | 0.59 | 4.08 | 0.11 | 0.14 | 8.25 | 0.94 |

**Relationship between uncertainty and amount of training data.** Next we investigate how aleatoric and epistemic uncertainty estimates change with varying amount of training data, under three different configurations with 1/2/3 weeks of training data and 2 weeks of test data on SimLA. Table 3 reports the average aleatoric uncertainty (AU) and epistemic uncertainty (EU) across test samples for each feature. We observe that ($i$) AU remains almost constant as training data size increases, which empirically shows that AU accounts for the inherent randomness/stochasticity in the data that is not a matter of the amount of training data, and that ($ii$) EU decreases noticably with increasing data size, suggesting that it can be reduced with sufficient training data.

**Relationship between EU and accuracy of each POI class**. From the results in Figure 5, we have the following observations: ($i$) As before, there is an inverse trend where accuracy decreases as epistemic uncertainty (EU) increases. This suggests that as the model becomes less certain (higher EU), its performance in predicting the correct POI class deteriorates. ($ii$) A higher frequency of POI classes (such as home and office buildings) generally leads to lower EU. Since EU aligns with amount of training data as showcased earlier, the more frequent POI classes, for which the model is trained on more observations, associate with lower EU and higher prediction accuracy. ($iii$) A lower frequency does not always mean high EU however. For example, POI:education in Figure 5 does not have a frequency as high as POI:home and POI:store with similar accuracies, but it has a relatively low estimated EU possibly because people's visits to such POI are highly patterned.

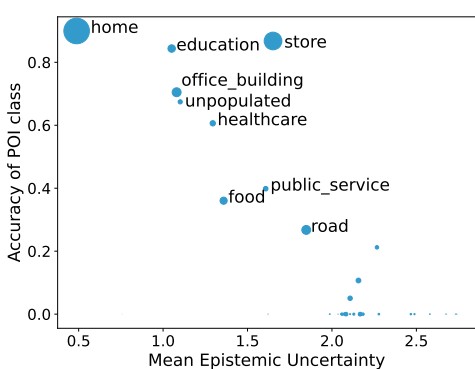

Figure 5: The relationship between epistemic uncertainty (EU) and accuracy of each POI class. Size of each point/POI is proportional to its frequency in the dataset. We only annotate a few POI labels for better illustration.

### 4.4 RQ4: ABLATION STUDY

As uncertainty is a key component of our anomaly detection model, we conduct ablation studies on its role from both the training and inference perspectives.

**Uncertainty from the training perspective**. To study the role of aleatoric versus epistemic uncertainty estimation, we consider three variants of UIFORMER: ($i$) UIFORMER-AU, which only implements the aleatoric uncertainty while keeping the model backbone the same as UIFORMER, ($ii$) UIFORMER-EU, which only models epistemic uncertainty; and ($iii$) UIFORMER-none, which only has the bare Dual Transformer and does not model epistemic or aleatoric uncertainty.

For comparison, we plot the relationship between uncertainty and performance in Figure 4. Firstly, the inclusion of uncertainty modeling can significantly enhance the predictive accuracy of numerical features. Moreover, UIFORMER-AU model significantly outperforms UIFORMER-EU in regression tasks. This suggests that aleatoric uncertainty captures essential variance within the dataset that epistemic uncertainty cannot address effectively. However, when considering classification tasks, such as POI and DOW, modeling aleatorical uncertainty does not provide a significant performance boost. UIFORMER slightly outperforms UIFORMER-AU, which suggests that modeling both aleatoric uncertainty and epistemic uncertainty can further improve the performance.

**Uncertainty from the inference/anomaly scoring perspective**. In this section, we investigate the impact of different anomaly scoring functions on detection performance.

Table 4 presents the results of UIFORMER's event-level anomaly detection using three different scores per feature: $\Delta_f$, pth($\Delta_f$), and pth $\left( \frac{\Delta_f}{1 + \alpha_f + \beta_f} \right)$. Here, $\Delta_f$ represents the feature delta error,

pth$(\cdot)$ denotes the percentile function, and $\alpha_f$ and $\beta_f$ represent aleatoric and epistemic uncertainties, respectively. The feature anomaly scores were aggregated by taking the maximum across features.

From the comparison between $\Delta_f$ and pth$(\Delta_f)$, we observe a significant improvement in performance when using percentiles. This shows that the percentile function and thus using the relative ranking of errors is beneficial when features are variable in scale, ensuring that features with smaller error scales are not overshad-

Table 4: Comparison of different anomaly scores

| Function | SimLA (small) | | NUMOSIM (small) | | NUMOSIM (20K) | |
|---|---|---|---|---|---|---|
| | AUROC | AUPR | AUROC | AUPR | AUROC | AUPR |
| $\Delta_f$ | 0.666 | 0.200 | 0.553 | 0.062 | 0.556 | 0.003 |
| pth$(\Delta_f)$ | 0.789 | 0.295 | **0.694** | 0.143 | 0.699 | 0.031 |
| pth$\left(\frac{\Delta_f}{1+\alpha_f+\beta_f}\right)$ | **0.795** | **0.329** | **0.694** | **0.147** | **0.700** | **0.034** |

owed by those with larger scales, leading to more balanced and effective anomaly detection. Furthermore, comparing pth$(\Delta_f)$ to pth$\left(\frac{\Delta_f}{1+\alpha_f+\beta_f}\right)$, we see that incorporating uncertainties into the anomaly scoring further enhances detection performance. This shows that accounting for total uncertainty improves the model's ability to identify the true anomalies more effectively.

## 5 RELATED WORK

**Human Mobility Modeling.** Methods in this field can be broadly categorized into traditional statistical models and the more recent deep learning approaches. Traditional statistical approaches typically rely on specific functional forms such as Poisson or Hawkes processes to predict event arrival times (Hawkes, 1971; Daley and Vere-Jones, 2008; Ogata, 1998), which struggle to capture the intricate spatiotemporal patterns present in human mobility data. Other statistical methods like Markov Chains, use transition matrices to predict future locations (Chen et al., 2014; Gambs et al., 2012; Cheng et al., 2013; Monreale et al., 2009) yet they fall short in modeling long-term dependencies. With the rise of deep learning, models such as RNNs and LSTMs have been used to learn the complex transition patterns (Gao et al., 2017; Song et al., 2016; Du et al., 2016), leading to better performance in the next location and trajectory prediction. Most recently, there has been a growing trend of using Transformer-based models for human mobility and trajectory data (Wan et al., 2021; Feng et al., 2018; Xue et al., 2021; Abideen et al., 2021; Wu et al., 2020; Wang and Osaragi, 2024), mainly inspired by its immense success in modeling inherently sequential data in NLP. However, most Transformer-based models overlook the underlying uncertainty in the data, especially as it pertains to human behavior over time. Specifically, most prior work focus on pretraining solely based on the masked reconstruction loss, without uncertainty estimation of the prediction. This motivates us to develop a new uncertainty-aware model to fill the gap, toward more effectively capturing human behavior that is often dependent on time and context.

**Uncertainty Estimation in Deep Neural Networks.** Uncertainty estimation has been studied in the field of NLP (Gal and Ghahramani, 2016b; Xiao and Wang, 2019) and CV (Kendall et al., 2016; Huang et al., 2018), which shows great promise in reflecting the underlying uncertainty in outcomes and thus improving task performance. One common approach is the Monte Carlo (MC) Dropout (Gal and Ghahramani, 2016a), which estimates uncertainty by applying dropout during inference, thereby improving model robustness. Others employ deep ensembles for model uncertainty estimation Lakshminarayanan et al. (2017). Kendall and Gal (2017) propose to learn aleatoric uncertainty through loss attenuation while modeling epistemic uncertainty using MC Dropout in both regression and classification tasks for CV. Mobiny et al. (2021) drop individual weights of the network during training and inference to measure uncertainty robustly. Motivated by the success of uncertainty estimation in these domains, we explore the potential benefit of uncertainty estimation in modeling human mobility, which forms the basis of our proposed model.

## 6 CONCLUSION

We introduced UIFORMER for human mobility modeling and anomalous behavior detection. UIFORMER is equipped with both data and model uncertainty estimation to account for the stochasticity inherent to human nature as well as the data scarcity in sufficiently observing complex human behavior. While its dual-attention mechanism captures dependencies among features and events, uncertainty estimation lends itself to more robust training and more nuanced anomaly scoring. Experiments on two benchmark datasets showed that UIFORMER outperforms existing baselines, while ablation analyses underscored the benefit of uncertainty-aware forecasting and anomaly detection.

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

## A    DATA STATISTICS

Table 5: Dataset statistics.

|  | SimLA | NUMOSIM |
|---|---|---|
| #Agents | 20,000 | 20,000 |
| #Anomalous Agents | 162 | 381 |
| #Events | 3,111,712 | 3,428,714 |
| #Anomalous Events | 892 | 3,468 |
| x range (km) | [-57, 57] | [-59, 59] |
| y range (km) | [-40, 40] | [-60, 60] |
| Avg of stay duration (min) | 509 | 455 |
| Avg of start time (min) | 850 | 776 |

## B    DETAILS OF TRANSFORMER BLOCK

**Transformer Block**, which contains two key components: a Multi-Head Self-Attention (MSA) layer and a Feed-Forward Network (FFN) layer. The MHA layer facilitates message passing between input tokens, while the FFN applies non-linear transformations to enhance feature extraction across different dimensions of the input vectors. To capture more complex interactions between tokens, multi-head attention is employed, where the attention mechanism is defined in Equation (11).

$$\text{Attention}(\boldsymbol{Q}, \boldsymbol{K}, \boldsymbol{V}) = \text{softmax}\left(\frac{\boldsymbol{Q}\boldsymbol{K}^T}{\sqrt{d}}\right)\boldsymbol{V}, \tag{11}$$

where $\boldsymbol{Q} \in \mathbb{R}^{N \times D}$, $\boldsymbol{K} \in \mathbb{R}^{N \times D}$, and $\boldsymbol{V} \in \mathbb{R}^{N \times D}$ represent the query, key, and value matrices, respectively, all projected from the same input matrix $\boldsymbol{E}$ (which have different forms in Feature-level Transformer and Event-level Transformer) with different learnable weight matrix. The softmax function transforms the scaled dot product into attention weights for $\boldsymbol{V}$, and $d$ is the dimensionality of $\boldsymbol{K}$ used for scaling the inner product. Besides, more high-order mutual information can be captured by stacking multiple Transformer blocks. Denote the embedding outputted by block $m \in \{1, \cdots, M\}$ as $e_i^m$, its updating process can be formulated as follows

$$\begin{aligned} \hat{e}_i &= e_i^{m-1} + \text{MSA}_i^m\left(\text{head}_1^{m-1}, \ldots, \text{head}_{n_{\text{head}}}^{m-1}\right) \\ e_i^m &= \hat{e}_i + \text{FFN}(\hat{e}_i). \end{aligned} \tag{12}$$

where $n_{\text{head}}$ is the number of heads.

## C    DETAILED SETTINGS FOR MASKED PREDICTION

*Baselines.* The following methods are chosen as baselines: (1) MLP (Murtagh, 1991), which applies multiple Linear layers to encode the event sequence, and decodes the masked features according to the embedding from the last layer. (2) LSTM (Hochreiter and Schmidhuber, 1997), a recurrent neural network commonly used to model sequential data, it reads the event sequence and decodes the masked event based on its hidden vector. (3) Transformer(Vaswani et al., 2023), one of the most powerful sequence models nowadays, which models the correlation of different events by multi-head self-attention. (4) Dual-Transformer, a variant of our proposed model, which removes the uncertainty learning mechanism and only keeps the dual Transformer architecture.

*Metrics.* We consider the prediction of numerical features and categorical features as regression and classification tasks, respectively. To this end, we apply MAE and MAPE to evaluate the performance of regression, and Accuracy (ACC) is applied to evaluate the classification performance. The unit for x, y are kilometers while the unit for start time and stay duration are minutes.

*Settings.* The models are trained on 1 GPU of NVIDIA RTX A6000. For all deep models, we search the parameters in a given parameter space, and each model's best performance is reported in the experiment. Here we report the parameter search space of our model: the mask ratio for pre-training is searched from [0.05, 0.3]. The embedding size for the transformer is searched from [32, 64, 128], and the layers for the transformer are searched from [2, 3, 5]. The batch size of each epoch is searched from 128 and the learning rate of Adam optimizer starts from 1e-3 with a weight decay 1e-05.

Table 6: Dataset statistics for anomaly detection. Ratio of anomalies depict the base rate.

|  | SimLA (small) | NUMOSIM (small) | NUMOSIM (20K) |
|---|---|---|---|
| #Agents | 254 | 851 | 20,000 |
| #Anomalous Agents | 162 | 381 | 381 |
| Ratio of Anomalous Agents | 0.638 | 0.448 | 0.019 |
| #Events | 18,963 | 73,129 | 1,706,468 |
| #Anomalous Events | 892 | 3,468 | 3,468 |
| Ratio of Anomalous Events | 0.041 | 0.047 | 0.002 |

## D  DETAILED SETTINGS FOR ANOMALY DETECTION

*Baselines.* We selected the following baseline methods for comparison: RioBus Bessa et al. (2016), Spatial-Temporal Outlier Detector (STOD) Cruz and Barbosa (2020), a self-supervised MLP (SS-MLP), LSTM, and Transformer. RioBus Bessa et al. (2016) is initially applied to bus trajectories and employs a Convolutional Neural Network (CNN) to predict the ID of the bus based on the sequence of GPS coordinates. For agent-level anomaly scores, it uses the negative probability of the correct prediction of the agent (bus) ID. For the stay-point (or event) level, scores are derived from the anomaly score of the 20-coordinate sub-trajectory ending at the given stay point. STOD Cruz and Barbosa (2020) is also originally applied to bus trajectories and is GRU-based to predict the bus ID. Its anomaly score computation approaches are similar to RioBus. SS-MLP takes event features including GPS coordinates and time as input. These features are first converted into embeddings, which are then concatenated to form a unified input vector. This vector is processed by an 8-layer MLP that outputs a probability indicating whether an event is real (1) or fake (0). During training, fake events are generated by randomly altering real events, and the model learns to classify between real and fake using binary cross-entropy loss. This approach allows the model to distinguish between normal and anomalous data. Finally, the negative of the output probability is used as the anomaly score, with a higher score indicating greater anomaly likelihood.

*Settings.* Since the percentage of anomalous events in the entire dataset is extremely small, we construct two additional test datasets with approximately 0.05% anomalous events, named SimLA (small) and NUMOSIM (small). These datasets are constructed by including all agents with anomalous events and randomly sampling from other agents without any anomalous events. The statistics of the test datasets are in Table 6. This approach provides three test datasets with varying levels of anomaly scarcity, allowing for diverse evaluation settings.

## E  CASE STUDY

Here, we present two examples: one agent with high total uncertainty and another with low uncertainty. Their daily trajectories in the first week of training data are shown in Figure 6. Markers of different shapes and colors represent various POI types. The numerical labels next to the markers indicate the order of events occurring each day. The high-uncertainty agent visits a wide variety of POI types daily, without any fixed pattern in the locations visited or the order of visiting those locations. In contrast, the low-uncertainty agent exhibits regular patterns on weekdays, such as traveling from home to a possible workplace, and remains at home all day on Sunday during the weekend. These examples demonstrate that our model's uncertainty measure effectively captures the variability in human mobility patterns.

## F  BROADER IMPACT

Our work proposes an unsupervised approach to modeling uncertainty in sequence data, with a specific focus on human mobility. By leveraging both aleatoric and epistemic uncertainties, our model captures the variability inherent in complex behaviors and provides a robust way to detect anomalies without relying on labeled data. This approach is highly generalizable and can be applied across various domains, such as finance (e.g., detecting credit card fraud or money laundering), online shopping (e.g., analyzing user purchasing patterns and identifying bot activity), and app user behavior analysis (e.g., detecting shifts or unusual engagement). By modeling uncertainty, the proposed

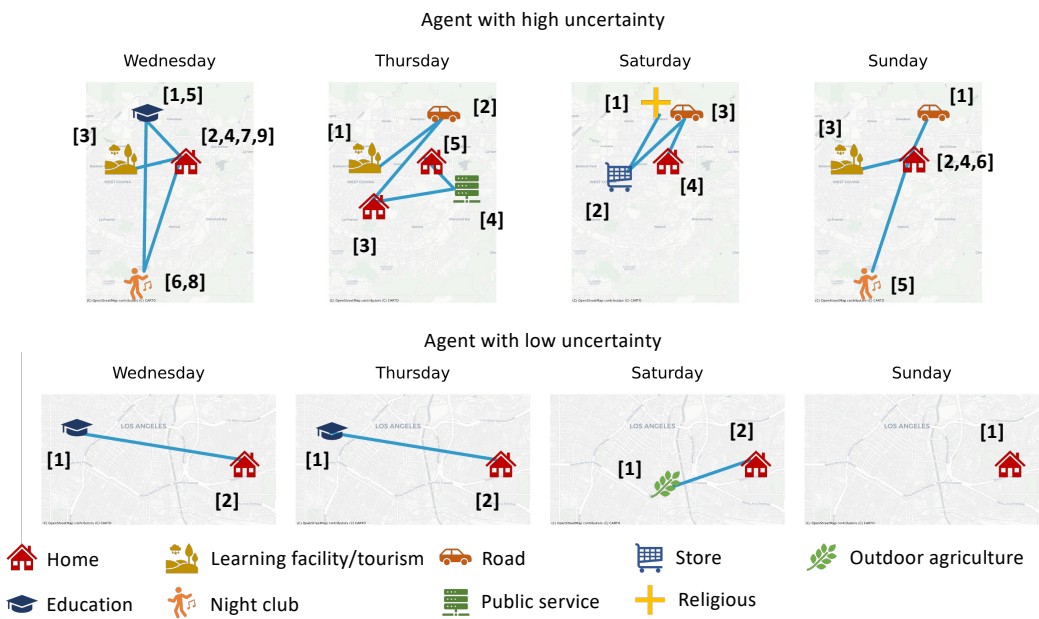

Figure 6: Case Studies of High-uncertainty and Low-uncertainty Agents

method can provide more informative anomaly scores, distinguishing between natural variability and significant deviations that require attention. This makes the model particularly useful in situations where uncertainty is expected, and traditional methods might lead to false positives.

