# OpenReview forum: "Uncertainty-aware Human Mobility Modeling and Anomaly Detection"
_ICLR.cc/2025/Conference — Submitted to ICLR 2025_

### Official Review · Reviewer_AKC3 · 2024-10-27

**Soundness:** 3
**Presentation:** 4
**Contribution:** 3
**Rating:** 6
**Confidence:** 5

**Summary:**

The paper proposes a model for detecting anomalies in human mobility using GPS data without labeled training data. It represents GPS data as sequences of stay-point events and utilizes Transformers for un/self-supervised learning. The model incorporates aleatoric and epistemic uncertainties to handle behavioral randomness and data sparsity, enabling robust training and decision-making. Experiments demonstrate its effectiveness over traditional forecasting and anomaly detection methods.

**Strengths:**

**S1.** The paper introduces a novel and practical approach to model both aleatoric and epistemic uncertainties in trajectories, allowing the model to handle the inherent randomness of human behavior and data sparsity effectively.

**S2.** Overall, the paper is technically sound and provides valuable contributions on how to deal with aleatoric and epistemic uncertainties with numerical features and categorical features together.

**S3.** The comprehensive experiments show the effectiveness of uncertainty modeling and ad performance.

**Weaknesses:**

**W1.** It would be better to provide more case studies to intuitively show the learned AU and EU.

**W2.** It would be better to include an addition experiment to show how $\lambda$ is selected in the loss function.

**W3.** It would be more interesting to explain why the authors chose their current set of baselines. More baselines are required. For example, it would be interesting to compare the proposed method with the existing time series outlier detection methods [1]. It would help the community to see whether existing methods like the Anomaly Transformer will work for human mobility anomaly detection.

_[1]. Anomaly Transformer: Time Series Anomaly Detection with Association Discrepancy, ICLR 2022._

**W4.** It would be better to include a section to introduce anomaly detection in sequential data (e.g., trajectory and time series) with more additional references as follows,

_[2]. Mobile Trajectory Anomaly Detection: Taxonomy, Methodology, Challenges, and Directions, IoTJ 2024._

_[3]. Time-series anomaly detection service at Microsoft, SIGKDD 2019._

_[4]. Anomaly detection in time series: a comprehensive evaluation, PVLDB 2022._

_[5]. Unraveling the 'Anomaly'in time series anomaly detection: a self-supervised tri-domain solution, ICDE 2024._

_[6]. Revisiting VAE for Unsupervised Time Series Anomaly Detection: A Frequency Perspective, WWW 2024._

_[7]. PeFAD: A Parameter-Efficient Federated Framework for Time Series Anomaly Detection, SIGKDD 2024._

_[8]. Deep variational graph convolutional recurrent network for multivariate time series anomaly detection, ICML 2022._

In addition, it is suggested to discuss the key differences between general time series/trajectory anomaly detection and the studied human mobility anomaly detection.

**W5.** As efficiency is important in anomaly detection, especially for downstream real-world applications, it is encouraged to compare the theoretical space and time complexities of the proposed method and baselines. Additionally, it would be better to include an experiment to show the training time and inference time comparison.

**Questions:**

1. How $\lambda$ is selected in the loss function?

2. There are various time series and trajectory anomaly detection methods. It would be interesting to investigate if these methods can be used to address human mobility anomaly detection.

3. It would be better to provide a running example to show how the uncertainties can be captured by the proposed uncertainty-aware decoder.

In addition, please see the weaknesses for other questions.

---

> ### Author Response · Authors · 2024-11-21
>
> Thanks for your comments, we address your concerns from the following aspects:
>
> (1) **Experiment and explanation of how $\lambda$ is tuned/selected in the loss function**: In our paper, we manually decide $\lambda$, so that the loss of numerical features and discrete features can be in the same scale.
>
> (2) **More baselines**: We appreciate your recommendation to strengthen the baseline methods. In designing our baselines, we aimed to cover core models that serve as the foundational backbone of many state-of-the-art approaches in human activity sequence modeling. For example, we included Transformer as a baseline, which has been foundational to SOTA human activity/trajectory sequence methods like [1-4]. Our goal was to ensure that the baselines represent essential frameworks widely recognized for their effectiveness in this domain, thus providing a fair comparison to our approach. Nevertheless, we value your suggestion and will consider incorporating additional SOTA baselines to enrich the evaluation further.
>
>
> [1] Lin, Yan, et al. "Pre-training context and time aware location embeddings from spatial-temporal trajectories for user next location prediction.", AAAI, 2021.
>
> [2] Chen, Yile, et al. "Robust road network representation learning: When traffic patterns meet traveling semantics." CIKM, 2021.
>
> [3] Jiang, Jiawei, et al. "Self-supervised trajectory representation learning with temporal regularities and travel semantics." ICDE, 2023.
>
> [4] Yang, Sean Bin, et al. "Lightpath: Lightweight and scalable path representation learning.", KDD 2023.
>
>
> (3) **A section to introduce anomaly detection in sequential data (e.g., trajectory and time series)**: Thanks for the question. Trajectory anomaly detection methods cannot be used to solve our event ad task directly. Their settings are different. Trajectories anomaly detection aims to make the decision given the original and destination, while in our task, we do not specify a given OD pair.
>
>
> (4) **Theoretical complexity analysis**. Given a sequence of $L$ events, with each event associated with $F$ features. The time complexity in the intra-event attention layer is $F^2 \cdot d$, and the complexity in the inter-event attention layer is $L^2 \cdot d$. Let $T$ denote the sample times at the inference process,  the total complexity of the proposed model is $T(F^2 \cdot d + L^2 \cdot d)$. In comparison, the complexity of the vanilla transformer is $L^2 \cdot d$. In contrast to the vanilla transformer, our proposed model sacrifices the complexity slightly but enhances the model's capacity to detect feature correlations within a single event and introduces the capability to learn uncertainties. This trade-off is deemed appropriate as our focus is on offline anomaly detection, where real-time decision-making is not required.
>
> We hope that we have fully addressed your concerns, and we look forward to your further input and your support of the paper on the confidence score.

---

### Official Review · Reviewer_NmDz · 2024-10-29

**Soundness:** 1
**Presentation:** 2
**Contribution:** 1
**Rating:** 3
**Confidence:** 5

**Summary:**

This paper presents a new method for human mobility modelling and anomaly detection via uncertainty estimation. The proposed method consists of a dual transformer encoder and an uncertainty-aware decoder.

**Strengths:**

1.	The motivation of the paper is clearly presented, and the writing is easy to follow.
2.	The proposed method estimates both epistemic uncertainty and aleatoric uncertainty.
3.	Both feature-level and event-level information are taken into account.
4.	An uncertainty-based score is proposed for anomaly detection.

**Weaknesses:**

1.	My main concern with the paper is the lack of novelty. The proposed method is merely the combination of existing techniques. The authors should explore more advanced techniques in the field. For example, the epistemic uncertainty is estimated by MC dropout which is a weight-space method. It can also be studied from a function-space perspective, using methods such as Gaussian processes, Hamiltonian Monte Carlo, Stein variational gradient descent, etc. As for aleatoric uncertainty, methods such as quantile regression and conformal inference are worth exploring as well.

2.	Another major issue of the paper is that the evaluation is not convincing. the baselines used for comparison in the experiments do not reflect the current state-of-the-art methods for human mobility modelling or uncertainty estimation.

Please refer to the relevant works listed below.
1) Abdar, Moloud, Farhad Pourpanah, Sadiq Hussain, Dana Rezazadegan, Li Liu, Mohammad Ghavamzadeh, Paul Fieguth et al. "A review of uncertainty quantification in deep learning: Techniques, applications and challenges." Information fusion 76 (2021): 243-297.
2)Sun, Shengyang, Guodong Zhang, Jiaxin Shi, and Roger Grosse. "FUNCTIONAL VARIATIONAL BAYESIAN NEURAL NETWORKS." In International Conference on Learning Representations, 2019.
3) Maddox, Wesley J., Pavel Izmailov, Timur Garipov, Dmitry P. Vetrov, and Andrew Gordon Wilson. "A simple baseline for bayesian uncertainty in deep learning." Advances in neural information processing systems 32 (2019).
4) Angelopoulos, Anastasios N., and Stephen Bates. "Conformal prediction: A gentle introduction." Foundations and Trends® in Machine Learning 16, no. 4 (2023): 494-591.

3.	The two datasets used for evaluation are synthetic. The proposed method should also be tested on real-world datasets.

**Questions:**

1.	What are the detailed experimental settings of the baseline methods used for comparison?
2.	Why not use real-world datasets for evaluation?
3.	Can the proposed also use other anomaly scores apart from the one proposed in the paper, e.g., reconstruction error?

---

> ### Author Response · Authors · 2024-11-21
>
> Thanks for your comments, we address your concerns from the following aspects:
>
> (1) **Concerns on novelty**. Thanks for your comments. Our main contribution is not the uncertainty modeling technique itself; in fact, we contribute by being the first to model both the aleatoric and epistemic uncertainty in behavior sequence learning, which can be easily generated to other domains with both numerical and categorical features. We appreciate your recommendation to explore alternative methods. These approaches are indeed promising, and we leave it as future work to further enhance our uncertainty estimation.
>
>
> (2) **Details on the experimental setting of baseline methods**  For SS-MLP, we searched embedding dimensions of 512 and 1024 with architectures of 7, 9, and 10 layers. We experimented with two input feature sets: one with latitude, longitude, and start time, and another adding stay duration and POI. We pick the best result from the experiments to report in the comparison.  For STOD and RioBus, we used embedding sizes of 512 and 1024. For LSTM and Transformer settings, please refer to Appendix Section C, where we detail the parameter search space shared across all deep models. The best-performing configuration is reported in the comparison.
>
>
> (3) **Experiment on real-world datasets**. Thanks for the advice.  There is no publicly available real human mobility dataset with known anomaly labels. Even for real human mobility data, it would be infeasible to manually inspect and label anomalies due to the complexity and scale of the data.
>
>
> (4) **Other anomaly scoring methods**.  Our anomaly scoring method computes the deviation between the predicted and true values for each event feature, which is similar to reconstruction error, as it reflects the difference between input features and their predicted (or reconstructed) values.  We have already experimented with three different scoring methods in Section 4.4.
>
> We hope that we have addressed your concerns, and we are looking forward to your further input and your support of the paper in score. Thank you.

---

> > ### Comment · Reviewer_NmDz · 2024-11-22
> > **Reply**
> >
> > The uncertainty modeling techniques have to be the primary focus for this work. Moreover, please dot not conduct research in a convenient manner, particularly if you aim to submit your work to a prestigious conference like ICLR. Based on the content presented in the paper, the theoretical understanding of uncertainty appears to be quite shallow. Frankly, I think this work barely scratches the surface of the current state-of-the-art.

---

### Official Review · Reviewer_ZgTj · 2024-10-29

**Soundness:** 2
**Presentation:** 3
**Contribution:** 1
**Rating:** 3
**Confidence:** 4

**Summary:**

This paper proposes an approach for anomaly detection of human mobility behaviour based on GPS data. The proposed approach is based on a Transformer architecture to model sequences of events/activities and it explicitly attempts to model epistemic and aleatoric uncertainty, which are then leveraged to devise a scoring function for rating abnormal behaviours. Using two simulated human activity datasets, the authors show that the proposed approach outperforms simpler baselines in modelling human mobility sequences and in detecting anomalous behaviors, especially showcasing and highlighting the contribution of modelling epistemic and aleatoric uncertainty to the results.

**Strengths:**

1) The problem of modeling human mobility data is an important one, with **important real-world implications**.
2) The overall approach proposed for this particular problem is **a unique and new combination of existing ideas from literature**. It is of significance for the urban mobility domain. Especially the focus on modeling uncertainty when modeling human mobility data is something that can be very relevant for practical applications but is typically ignored in the urban mobility literature.
3) The paper is **very well written**. The approach is mostly presented in a **clear** way (with some expectations, as mentioned below) and, in most cases, with a sufficient level of detail.
4) The **empirical results show significant improvements** over the simpler baselines considered.

**Weaknesses:**

1) Although the specific combination of existing ideas in the proposed approach is new from the application point of view, from a methodological point of view, the **low novelty** is a concern, especially **for the broader ICLR audience**. Since the contribution is of significance mostly for a transportation audience, I would encourage the authors to consider targeting a transportation journal instead since I believe this work would have more visibility and more impact there. Otherwise, for ICLR, the authors should motivate better the broader applicability of their work and their insights beyond the concrete application considered. For example, could the uncertainty-aware approach be applied to other sequence modeling tasks in different domains? Could you give some examples?
2) The **motivation for the application considered should be improved**. Why would one want to detect abnormal mobility behaviours? The authors briefly mention that they can "indicate security threats or spread of infectious disease", but these use cases need to be better explained and backed up by related works where having such predictions of abnormal mobility behaviours would be relevant. This is the critical basis for the proposed work, so it needs to be articulated in a more convincing way.
3) The **baselines considered in the paper are very naive**. However, modelling sequences of human activities is a very well-studied problem in the transportation field, and the literature is vast on different deep learning approaches, including approaches based on transformers. I encourage the authors to consider including stronger baselines from the state-of-the-art for modelling human activity sequences - i.e., at least for the predictive part (RQ1); although the extension of some of these for score-based anomaly detection is also trivial, and therefore can also be potentially considered as additional baselines for RQ2.
4) The **soundness of the experimental setup is a concern**, given that the datasets used for validating the proposed approach are themselves simulated based on deep learning models that were trained on real-world data to generate synthetic sequences. The authors are essentially fitting a deep learning model to the output of another deep learning model. This raises at least three important questions: i) is the output of the deep generative model good enough to generate realistic synthetic sequences that capture well the data distribution, including the "long-tails" such that abnormal behaviors are captured? and ii) why not simply use the deep generative model that was used to generate the data to detect abnormal behaviors? What are the advantages of using the proposed transformer architecture instead? iii) why haven't the authors considered using real data to demonstrate their approach (of course, despite a quantitative evaluation being difficult/impossible)? I would suggest the authors discuss these concerns and discuss their rationale for using simulated data and the potential limitations or advantages of this approach in the paper, as well as discuss the challenges and potential approaches for evaluating real-world data.
5) There are several aspects of the proposed approach that are either **not properly motivated or unclear**. For example, eq. 7 is presented as "the average predicted logits"; however, looking at eq. 7, it seems to be the average of the exponentiated logits (i.e., likelihoods), and then the authors take the logarithm. This should be clarified and properly motivated. Also, the authors have multiple passes T for the categorical loss in eq. 7, but not for the numerical features loss in Section 3.3.2. What is the reasoning behind this distinction in approaches? Similarly, what is the reasoning for the +1 in eq. 10? I would suggest the authors provide a more detailed explanation or derivation for equation 7, and explicitly state their reasoning for the different approaches used for categorical and numerical features, as well as the addition of 1 in equation 10.
6) Lastly, there are **several inconsistencies** in the paper. For example, it is written that "MAPE for SD from 51.95% to 21.93%", however these numbers don't match the values in Table 1. Similarly, it is mentioned that "UIFORMER delivers the best performance across both datasets and for all features compared with the baselines.". However, the MAPE of UIFORMER for ST is significantly worse than a simple transformer. These aspects should be clarified in the paper. It is also written that "AU remains almost constant as training data size increases"; however, looking at the table there are several significant exceptions that should be considered and discussed. I suggest that the authors carefully review their results and claims for accuracy, and provide a more nuanced discussion of their findings, particularly where the results don't align perfectly with their general claims.

**Questions:**

Kindly see my question in the weaknesses section, especially the questions that I raised in bullets 2, 4 and 5 above.

---

> ### Author Response · Authors · 2024-11-21
> **Part 1**
>
> Thanks for such detailed and constructive comments! we address your concerns in the following points:
>
> (1) **Broader impact of our work**. Thanks for raising this question. We would like to argue that our paper does not only target at transportation audience. Indeed, our method can be easily generated to sequence modeling tasks in many other domains. We kindly directly you to the "Broader Impact" section in the appendix: "This approach is highly generalizable and can be applied across various domains, such as finance (e.g., detecting credit card fraud or money laundering), online shopping (e.g., analyzing user purchasing patterns and identifying bot activity), and app user behavior analysis (e.g., detecting shifts or unusual engagement)."
>
> (2) **Motivation of abnormal mobility behavior detection**. The motivation for detecting abnormal mobility behaviors lies in the potential to address critical societal needs in public health, safety, and urban management. For example, during a pandemic, detecting anomalies helps monitor social distancing and identify potential hotspots, aiding health officials in preventing the spread of infectious diseases [1,2]. Similarly, unusual movement patterns can signal emergencies like natural disasters and extreme weathers, allowing for faster, more targeted responses that enhance urban resilience [3]. Beyond these, abnormal mobility detection supports crowd management and public safety by identifying patterns that may indicate overcrowding or potential security threats [4]. In urban planning, detecting these anomalies allows city planners to better understand evolving trends in transportation which can help optimize infrastructure and improve public transit services [5]. Each of these applications demonstrates the importance of human mobility anomaly detection.
>
>
> [1] Shearston, Jenni, et al. "Social-distancing fatigue: Evidence from real-time crowd-sourced traffic data", Sci Total Environ, 2021.
>
> [2] Nakamoto, Daisuke, et al. "The impact of declaring the state of emergency on human mobility during COVID-19 pandemic in Japan",  Clin Epidemiol Glob Health, 2022.
>
> [3] Li, Xiang, et al. "Using human mobility data to detect evacuation patterns in hurricane Ian", Annals of GIS, 2024.
>
> [4] Sanchez, Thomas, et al. "The Implications of Human Mobility and Accessibility for Transportation and Livable Cities", Urban Science, 2023.
>
> [5] Shuo, Shang, et al. "Human Mobility Prediction and Unobstructed Route Planning in Public Transport Networks", MDM, 2014.
>
> (3) **Concerns on baselines**. We appreciate your recommendation to strengthen the baseline methods. In designing our baselines, we aimed to cover core models that serve as the foundational backbone of many state-of-the-art approaches in human activity sequence modeling. For example, we included Transformer as a baseline, which has been foundational to SOTA human activity/trajectory sequence methods like [1-4]. Our goal was to ensure that the baselines represent essential frameworks widely recognized for their effectiveness in this domain, thus providing a fair comparison to our approach. Nevertheless, we value your suggestion and will consider incorporating additional SOTA baselines to enrich the evaluation further.
>
>
> [1] Lin, Yan, et al. "Pre-training context and time aware location embeddings from spatial-temporal trajectories for user next location prediction.", AAAI, 2021.
>
> [2] Chen, Yile, et al. "Robust road network representation learning: When traffic patterns meet traveling semantics." CIKM, 2021.
>
> [3] Jiang, Jiawei, et al. "Self-supervised trajectory representation learning with temporal regularities and travel semantics." ICDE, 2023.
>
> [4] Yang, Sean Bin, et al. "Lightpath: Lightweight and scalable path representation learning.", KDD 2023.

---

> ### Author Response · Authors · 2024-11-21
> **Part 2**
>
> (4) **Concerns on Experiment Setup**. We'd like to address your concerns from the following points: Firstly, we did not have access to the original simulation model itself and we could only access the simulated dataset it produced. This means we were unable to use the generative model directly for anomaly detection. Furthermore, the purpose of the simulation model was solely to generate synthetic data for research purposes, rather than for anomaly detection tasks.  Secondly, the generation model was learned on real samples from 2017 National Household Travel Survey (NHTS) [1] and anomalies are specifically designed and injected to reflect realistic abnormal behaviors in human mobility. This setup aims to capture a wide range of patterns, including long-tail distributions, to imitate real-world scenarios as closely as possible. Thirdly, there is no publicly available real human mobility dataset with known anomaly labels. Even for available human mobility datasets, manually labeling anomalies would be impractical due to the scale and complexity of the data.
>
> [1] Stanford, Chris, et al. "NUMOSIM: A Synthetic Mobility Dataset with Anomaly Detection Benchmarks." Proceedings of the 1st ACM SIGSPATIAL International Workshop on Geospatial Anomaly Detection. 2024.
>
> (5) [R2Q5]**Details on model design**. Thanks for the comments. The design of Eq.7 is mainly inspired by [1]. In the paper, we present Eq.7  as "average predicted logits, normalized over all possible categorical outcomes", which utilizes softmax and therefore introduces the exponential term. The reason for +1 in eq.10 is to let the anomaly score be more numerical stable in case $\alpha_f + \beta_f$ equals 0.
>
> Concern regarding multiple passes for the categorical loss but not for the numerical features. The AU of the numerical feature is modeled as a learnable loss attenuation term, which can be learned in a one-forward pass. However, to model the AU for categorical features, it is not trivial to use the same loss function as in numerical features. Following [1], for the categorical feature, we model the AU term $\sigma_f$ as the standard deviation of the logits (see Eq.6), which requires multiple passes to learn the AU term, so that we can penalize less when the data has high uncertainty.
>
> [1] Alex Kendall and Yarin Gal. 2017. What uncertainties do we need in Bayesian deep learning for
> computer vision?

---

> ### Author Response · Authors · 2024-11-21
>
> We hope that we have addressed your concerns, and we are looking forward to your further input and your support of the paper in score.

---

> > ### Comment · Reviewer_ZgTj · 2024-11-25
> >
> > I thank the author for the detailed response. It addresses some of my concerns. However, my key concerns regarding the low novelty work and broader applicability of the proposed approach remain, and in some sense, they were aggravated. In the response, it is mentioned that "our paper does not only target a transportation audience. Indeed, our method can be easily generated to sequence modeling tasks in many other domains". However, confusingly, in response to reviewer NmDz, it is mentioned that "Our main contribution is not the uncertainty modeling technique itself. In fact, we contribute by being the first to model both the aleatoric and epistemic uncertainty in behavior sequence learning, which can be easily generated to other domains with both numerical and categorical features". This is a domain-specific contribution (i.e. for behavior sequences). If the proposed approach can be applied to other domains, then the authors should consider broadening their evaluation to include datasets from other application domains as well and re-write the paper accordingly. Overall, the experiments should be aligned with the claimed novelty. For example, there is extensive literature on modelling "both the aleatoric and epistemic uncertainty". If this is a novel aspect that the authors want to highlight, then I would encourage them to better account for other alternative approaches from the SOTA, as reviewer NmDz also points out.
> > With this in mind, I will keep my original score, but I thank the authors again for their detailed responses.

---

### Official Review · Reviewer_McKL · 2024-10-29

**Soundness:** 3
**Presentation:** 3
**Contribution:** 3
**Rating:** 6
**Confidence:** 2

**Summary:**

The paper proposes UIFORMER, a dual-Transformer architecture for modeling human mobility patterns and detecting anomalies. It combines aleatoric and epistemic uncertainty estimation to better capture uncertainties inherent in human mobility data. The model aims to enhance robustness in anomaly detection by incorporating uncertainty into decision-making.

**Strengths:**

1. Originality: The model’s design, utilizing a dual-Transformer architecture for feature- and event-level attention, effectively captures dependencies at different levels of human behavior.
2. Quality: The paper includes comparisons with various baselines, showing that UIFORMER achieves superior performance for anomaly detection.
3. Clarity: The paper is well-organized and clearly explains the model’s architecture, training dynamics, and uncertainty estimation.  Figures and tables enhance the understanding of the model’s performance and design.
4. Significance: The method’s ability to better capture uncertainties in human behavior could have significant implications for applications like security, surveillance, and health monitoring, where accurate anomaly detection is critical.

**Weaknesses:**

1. Lack of Novelty in Uncertainty Quantification: The techniques for quantifying aleatoric and epistemic uncertainty, including Monte Carlo (MC) Dropout, have been widely used in other domains. The absence of a discussion on related works, specifically evidential deep learning methods, limits the novelty of the approach. Incorporating such references and discussions could strengthen the contribution of this paper.
2. There is insufficient detail about the SimLA dataset, particularly whether it is open-source or publicly available. Clarifying this point would improve transparency and reproducibility.
3. The decision to mask only one event per sequence in validation and testing phases is restrictive and may not reflect the full potential of the model in real-world scenarios where multiple events may be missing.
4. The use of MC Dropout for epistemic uncertainty estimation limits the exploration of alternative methods. Testing more methods, such as deep ensembles or evidential neural networks, could provide a more comprehensive understanding of uncertainty estimation in this context.

**Questions:**

1. Could the authors elaborate on why only one event is masked per sequence during validation and testing? Would masking more events per sequence affect the performance or the uncertainty estimates?
2. Is the SimLA dataset open-sourced, and are there plans to release it publicly? This would be useful for reproducibility and benchmarking future models.
3. Why was Monte Carlo Dropout chosen for epistemic uncertainty estimation, and are there plans to experiment with other methods like deep ensembles or evidential neural networks?
4. Are there specific applications or downstream tasks, such as traffic signal adjustments or navigation, where the proposed model could be directly applied to demonstrate practical benefits?

---

> ### Author Response · Authors · 2024-11-21
>
> Thanks for the contrastive comments, we address your concern in the following points:
>
> (1) **Concerns on novelty in Uncertainty Quantification**. Thanks for your comments. Our main contribution is not the uncertainty modeling technique itself, in fact, we are the first to model both the aleatoric and epistemic uncertainty in human behavior sequence learning. MC Dropout is only a component of our work, which equips the model with epistemic uncertainty learning ability.
>
> (2) **Details about SimLA**. The SimLA dataset is a private dataset, and there are currently no plans for public release. It is generated using the same simulation process as the publicly available NUMOSIM dataset.
>
> (3) **Reason of choosing MC Dropout for EU**. We chose Monte Carlo Dropout for epistemic uncertainty estimation because (1) it is a simple yet effective approach that has been extensively validated across various domains; (2) it can be seamlessly incorporated with aleatoric uncertainty modeling. (3) it is a practical choice for our initial experiments while maintaining strong empirical performance. We appreciate your recommendation to explore alternative methods, such as deep ensembles and evidential neural networks. These approaches are indeed promising, and we leave it as future work to further enhance our uncertainty estimation.
>
> (4) **Downstream tasks**. There are multiple downstream tasks that the proposed model can be helpful.
>
> 1. Pandemic Monitoring and Public Health: During a pandemic, detecting mobility anomalies can help track compliance with social distancing and identify areas with unexpected gatherings, which helps health officials enforce measures and allocate resources. [1]
>
> 2. Urban Resilience to Emergencies: Anomalous movement patterns, such as sudden evacuations, can signal emergencies like natural disasters or extreme weather. Detection supports efficient response by directing resources to affected areas [2][3].
>
> 3. Urban Safety and Crowd Management: Identifying deviations in movement patterns can help detect public safety risks, including crowd congestion, unexpected gatherings, and potential terrorism. [4]
>
> 4. Transportation Planning: Anomalies in mobility patterns reveal issues like road closures or transit delays, enabling city planners to optimize routes, reduce congestion, and improve transit services. [5]
>
> [1] Shearston, Jenni, et al. "Social-distancing fatigue: Evidence from real-time crowd-sourced traffic data", Sci Total Environ, 2021.
>
> [2] Nakamoto, Daisuke, et al. "The impact of declaring the state of emergency on human mobility during COVID-19 pandemic in Japan",  Clin Epidemiol Glob Health, 2022.
>
> [3] Li, Xiang, et al. "Using human mobility data to detect evacuation patterns in hurricane Ian", Annals of GIS, 2024.
>
> [4] Sanchez, Thomas, et al. "The Implications of Human Mobility and Accessibility for Transportation and Livable Cities", Urban Science, 2023.
>
> [5] Shuo, Shang, et al. "Human Mobility Prediction and Unobstructed Route Planning in Public Transport Networks", MDM, 2014.

---

> > ### Comment · Reviewer_McKL · 2024-11-30
> > **Thanks for the feedback**
> >
> > Can the authors discuss the following baseline: adding evidential deep learning techniques [1,2] into an existing human behavior sequence learning model, how is the baseline different from the proposed method?
> >
> > [1] Sensoy, Murat, Lance Kaplan, and Melih Kandemir. "Evidential deep learning to quantify classification uncertainty." Advances in neural information processing systems 31 (2018).
> >
> > [2] Amini, Alexander, et al. "Deep evidential regression." Advances in neural information processing systems 33 (2020): 14927-14937.

---

### Meta-Review · Area_Chair_RvDR · 2024-12-24

**Metareview:**

While the submission presents an interesting and original application of uncertainty quantification for anomaly detection in human mobility data, several key issues must be addressed for it to be considered a strong contribution to the ICLR community:

The novelty of the approach is limited, as it relies on existing techniques for uncertainty estimation. Exploring alternative methods and providing a more comprehensive discussion of the state-of-the-art in uncertainty quantification would be beneficial. The use of synthetic datasets undermines the robustness of the experimental evaluation. Real-world datasets should be incorporated to strengthen the model's real-world applicability. The choice of baselines and the motivation behind the application of anomaly detection in mobility data need to be reconsidered and expanded upon.

Given the concerns about novelty, dataset choice, and experimental rigor, I recommend rejecting the paper in its current form.

**Additional Comments On Reviewer Discussion:**

Despite some improvements, there are still unresolved issues regarding the novelty of the approach, the thoroughness of the evaluation, and scalability. These concerns were acknowledged by the authors but not fully addressed in the rebuttal. The score remains unchanged by the reviewers.

---

### Decision · Program_Chairs · 2025-01-22

Reject